# A Bibliometric Analysis of Digital Literacy Research from 1990 to 2022 and Research on Emerging Themes during the COVID-19 Pandemic

Chen Wang * and Li Si *

School of Information Management, Wuhan University, Wuhan 430072, China
* Correspondence: simcwang@whu.edu.cn (C.W.); lsiwhu@163.com (L.S.)

**Abstract:** Due to the rapid advancement of digital technology and its contribution to sustainable development, digital literacy has become an increasingly significant research topic. However, the uneven distribution of new technologies has caused emerging inequalities, which have been exacerbated by the COVID-19 pandemic, one of the most significant public health crises of the century. This paper aims to conduct an in-depth analysis of scientific production using bibliometric methods to comprehend the current research status of digital literacy studies and evaluate the pandemic's impact on such research. In total, 7523 documents published between 1990 and 2022 were identified and analyzed using bibliometric research methods in the Web of Science database. These methods included growth trend analysis, network analysis, highly cited literature analysis, factor analysis, and time-series-based analysis of frequently discussed topics. Additionally, a word cloud analysis of the keywords in digital literacy literature from 2020 to 2022, during the COVID-19 pandemic, was created. The study's outcomes explore digital literacy research, including current trends, significant publications, and institutions involved in the field. This study emphasizes the importance of digital literacy in today's society, particularly during the COVID-19 pandemic. It also highlights the potential of bibliometric analysis as a tool for identifying research gaps.

**Keywords:** digital literacy; bibliometric analysis; COVID-19 pandemic



## 1. Introduction

The digital age has revolutionized the way in which we communicate, learn, and interact with each other. Digital literacy research has gained significant attention in recent years due to the widespread use of digital technologies. Understanding the various dimensions of digital literacy is essential for individuals, organizations, and society as a whole. Digital literacy is defined as the ability to use digital technology, communication tools, and networks to locate, evaluate, use, and create information [1]. It is a critical research topic in information science [2], relevant to various issues such as information overload, lifelong learning, knowledge management, and the development of the information society [3]. The concept of digital literacy is continuously evolving with advancements in digital technology. The emergence of the concept of digital literacy during the 1990s is widely acknowledged by various authors. Specifically, the term "digital literacy" was often used to describe the capacity to read and comprehend hypertext and multimedia texts. Lanham, R.A. (1995) posited that digital literacy was interchangeable with "multimedia literacy" [4]. Meanwhile, Glister, P. (1997) introduced the definition of digital literacy as the ability to comprehend and utilize information presented by computers in diverse formats from various sources [5]. Similarly, Pool, C.R. (1997) viewed digital literacy as an essential skill that involves the capability to comprehend, assess, and integrate information in numerous computer delivery formats [6]. Eshet, Y. (2004) argued that digital literacy goes beyond the mere operation of digital devices or software; it involves a range of complex cognitive, motor, sociological, and emotional skills that users need to function effectively in a digital environment [7]. To

meet these requirements, Eshet, Y proposed a conceptual framework for digital literacy that included five literacy skills: visual literacy, reproductive literacy, information literacy, branch literacy, and social–emotional literacy [8].

Similarly, Martin, A. (2005) defined digital literacy as "the awareness, attitude, and ability of individuals to appropriately use digital tools and facilities to identify, access, manage, integrate, evaluate, analyze, and synthesize digital resources; construct new knowledge; create media expressions; and communicate with others in the context of specific life situations, in order to enable constructive social action and to reflect upon this process" [9]. Digital literacy is viewed as a convergence of various literacies, including IT literacy, information literacy, technology literacy, media literacy, and visual literacy, as the relationships between these factors and the digital sphere become more apparent [10].

Meyers, E.M. (2013) suggested that the definition of digital literacy ranges from simple technical fluency to the application of information literacy skills, such as locating, extracting, organizing, managing, presenting, and evaluating information, in a digital environment. The concept of digital literacy is further expanded in a framework that encompasses various skills, understandings, norms, and practices [11]. Spante, M. (2018) defined digital literacy by referencing policy documents from the EU and OECD, emphasizing that digital literacy entails an individual's ability to live, learn, and work in a digital society while also having the power to make informed decisions and achieve goals [12]. Pangrazio, L. (2020) conducted a comparative review of publication pairs in three language contexts and found challenges in conceptualizing and applying the concept of digital literacy in all contexts [13]. Overall, the concept of digital literacy is constantly evolving with the advancement of information technology.

This study aims to provide a comprehensive overview of research trends, publication output, and emerging topics in digital literacy research, particularly during the COVID-19 pandemic, through a bibliometric analysis. Additionally, the study aims to demonstrate the potential of bibliometric analysis as a research tool.

The research seeks to answer the following five research questions.

1.  What are the global digital literacy research trends?
2.  Which authors and countries have actively contributed to digital literacy research?
3.  What are the most important cited articles that contribute to the body of knowledge in digital literacy research?
4.  What are the hot research topics and emerging trends in digital literacy research?
5.  What are the topics in digital literacy research that emerged after the COVID-19 pandemic?

To address these questions, we analyzed 7523 pieces of literature related to digital literacy published since the 1990s using bibliometric methods. We also analyzed the keyword content of 3961 articles published between 2020 and 2022 using word cloud graphs.

## 2. Review of the Relevant Literature

### 2.1. Review of Digital Literacy

A significant body of literature already exists on the advancement of digital literacy research. To efficiently track the progress of this research, we utilize Table 1 to organize the content of these documents.

In summary, the relevant review content of digital literacy research primarily focuses on the construction of digital literacy models, the digital divide, education, health, and digital security. The research objects include children, adolescents, the elderly, patients, students, and teachers, among others. The research methods primarily employ qualitative reviews of the literature.

**Table 1.** Review of digital literacy.

| Authors | Research Content |
| --- | --- |
| Pérez-Escoda, A et al. (2019) [14] | Dimensions of digital literacy based on five models of development. |
| Reddy, P. (2020) [15]<br>Reddy, P. et al. (2021) [16] | Digital literacy: a review of literature. |
| Caldevilla-Dominguez, D (2021), Reyes, C.E.G. (2021) and Neto, N.V.L.et al. (2022) [12,17–24] | The authors analyzed the literature on digital literacy in learning and education and found that, in terms of digital literacy, teachers and students showed great interest in global and multi-disciplinary fields. The key factors of digital literacy include digital literacy, digital learning, and digital skills in the 21st century. Students should develop digital literacy skills to meet the competency needs of the digital age. |
| Wang, X.X (2022), Palumbo, R. (2022,) and Choukou, M.A. et al. (2022) [25–29] | Summarizing the research on eHealth literacy and digital health literacy, it is believed that eHealth and digital health literacy intersect and that populations at risk of limited health literacy are equally vulnerable to digital health literacy challenges. However, the literature on the topic is sparse and immature, and health information literacy research needs further development. |
| Suslo, R. (2018), Oh, S.S. et al. (2021) [30,31]. | The authors conducted a systematic review of digital literacy research among the elderly. They believed that it is necessary to conduct more research on evaluating digital literacy among the elderly. It was thought that it is essential to carry out more research on the evaluation of digital literacy assessments for the elderly. It is necessary to pay attention to the cultivation of digital literacy among the elderly as a means of ensuring their health needs and human rights in the realities of the twenty-first century. |
| Gonzalez-de-Eusebio, J. et al. (2021) [32] | Ethology of media literacy in the twenty-first century's digital society: a literature review. |
| Choudhary, H. et al. (2022) [33] | Addressing the digital divide through digital literacy training programs: a systematic literature review. |
| Tinmaz. H. et al. (2022) [34] | A systematic review on digital literacy. |
| Peng, D.H. and Yu, Z.G. et al. (2022) [35] | A literature review of digital literacy over two decades. |
| Estrada, F.G.R. et al. (2022) [36] | Security as an emerging dimension of digital literacy for education: a systematic literature review. |
| Yu, Z.G. et al. (2022) [37]. | Sustaining student roles, digital literacy, learning achievements, and motivation in online learning environments during the COVID-19 pandemic. |

### 2.2. Digital Literacy Research during the COVID-19 Pandemic

Three years have passed since the end of 2019, and the COVID-19 pandemic has significantly impacted the research related to digital literacy. The relevant literature can be categorized by year, as the impact of the pandemic on digital literacy research evolves annually.

In 2020, Adnan, M. andAnwar, K. conducted a survey on the situation of online learning in higher education in Pakistan. They found that online learning cannot achieve ideal results in underdeveloped countries such as Pakistan [38]. Beaunoyer, E. et al. explored the interplay between the COVID-19 crisis and digital inequality and proposed effective solutions to address the crisis' dire consequences [39]. Jena, P.K. argued that the outbreak of COVID-19 has taught us that change is inevitable and has become a catalyst for educational institutions to develop and utilize technology platforms that were previously unused [39]. Nouri, S. argued that the coronavirus crisis exposed the differences in access to care among vulnerable groups, especially in the domain of telemedicine [40]. Ramsetty, A. predicted that the COVID-19 pandemic would alter the way we provide healthcare and that we would likely rely more on technology and integrate it further in the future [41]. Rashid, S. et al. suggested that the COVID-19 outbreak led to a downward spiral in the world economy and had a significant impact on the higher education system. The sudden closure of campuses

as a social distancing measure to prevent the virus's spread required higher education institutions and universities to plan post-pandemic education and research strategies to ensure student learning outcomes and standards of educational quality [42].

In 2021, Rodriguez, J.A. conducted a comparison of the use of telephone and video telemedicine access during the COVID-19 pandemic [43]. Sari, F.M. and fellow researchers found that an online learning platform is a useful tool for supporting an online learning environment [44], while Lischer, S. posited that the damage caused by COVID-19 to the education sector may last longer than originally predicted [45]. Patel, U. et al. investigated whether information acquisition, attitudes, and behaviors related to COVID-19 among American college students were associated with health literacy and digital health literacy [46].

In 2022, Tabroni, I. et al. aimed to implement and strengthen the literacy campaign in non-primary schools after COVID-19 [47]. Yu, Z. et al. conducted research to develop an online sustainable education model to promote this learning method [48]. Babel, H. et al. conducted a bibliometric analysis of emerging themes in digital literacy research before the COVID-19 pandemic [49]. Under the influence of COVID-19, digital literacy research in 2020 primarily focused on digital gaps and disadvantaged groups. In 2021, research on digital literacy shifted its focus towards remote medical care and online teaching.

## 3. Methodology

### 3.1. Bibliometric Analysis

Bibliometric analysis is a quantitative method used to determine the volume and growth patterns of literature in a particular emerging field. In this study, two effective techniques were utilized: performance analysis and scientific mapping. Performance analysis evaluated the publications' performance in terms of publication output by country, author, affiliation, and growth trends over the years. Scientific mapping techniques included citation analysis, co-citation analysis, bibliographic coupling, co-word analysis, and co-authorship analysis. When these methods were used in combination with network analysis, they were instrumental in presenting the bibliometric and intellectual structure of this research field. Bibliometric analysis is a useful and functional tool for deciphering and mapping the cumulative scientific knowledge and evolutionary nuances of well-established fields by making sense of large volumes of unstructured data in rigorous ways [50–52].

### 3.2. Data Source

Our literature review was based on the Web of Science (WoS), which indexes high-quality publications from top journals and ranked international conferences [53,54]. The essential databases of WoS are core collections that provide access to global academic information, covering over 21,800 high-impact academic journals with worldwide authority across various fields such as the natural sciences, engineering, technology, biomedicine, social sciences, the arts, and the humanities. The WoS' core collections have a strict screening mechanism that adheres to Bradford's Law in bibliometrics, which only includes critical academic journals in various disciplines. Additionally, the WoS' core collections comprise references cited in papers, and a unique citation index is compiled according to the cited authors, sources, and publication years, allowing users to trace the origin and history of a research document or track its recent progress [55].

For our study, we selected literature data included in the Web of Science (WoS) as the data source (11 December 2022). The retrieval dataset was the WoS core collection, and the retrieval subject was "digital literacy". As the concept of digital literacy began to emerge in the 1990s, we limited the publication date range to the period between "1 January 1990" and "31 December 2022". A total of 10,337 pieces of literature data were retrieved. To ensure the representativeness of the literature and the complexity of measurement, we selected 7523 documents, including literature papers (n = 7087, 94.2%) and review papers (n = 436, 5.8%). We derived records containing complete information, such as author, address, year of publication, source journal, title, subject category, and references, including the abstract

and cited references, from all 7523 documents. The stages of article retrieval and further analysis are displayed in Figure 1.

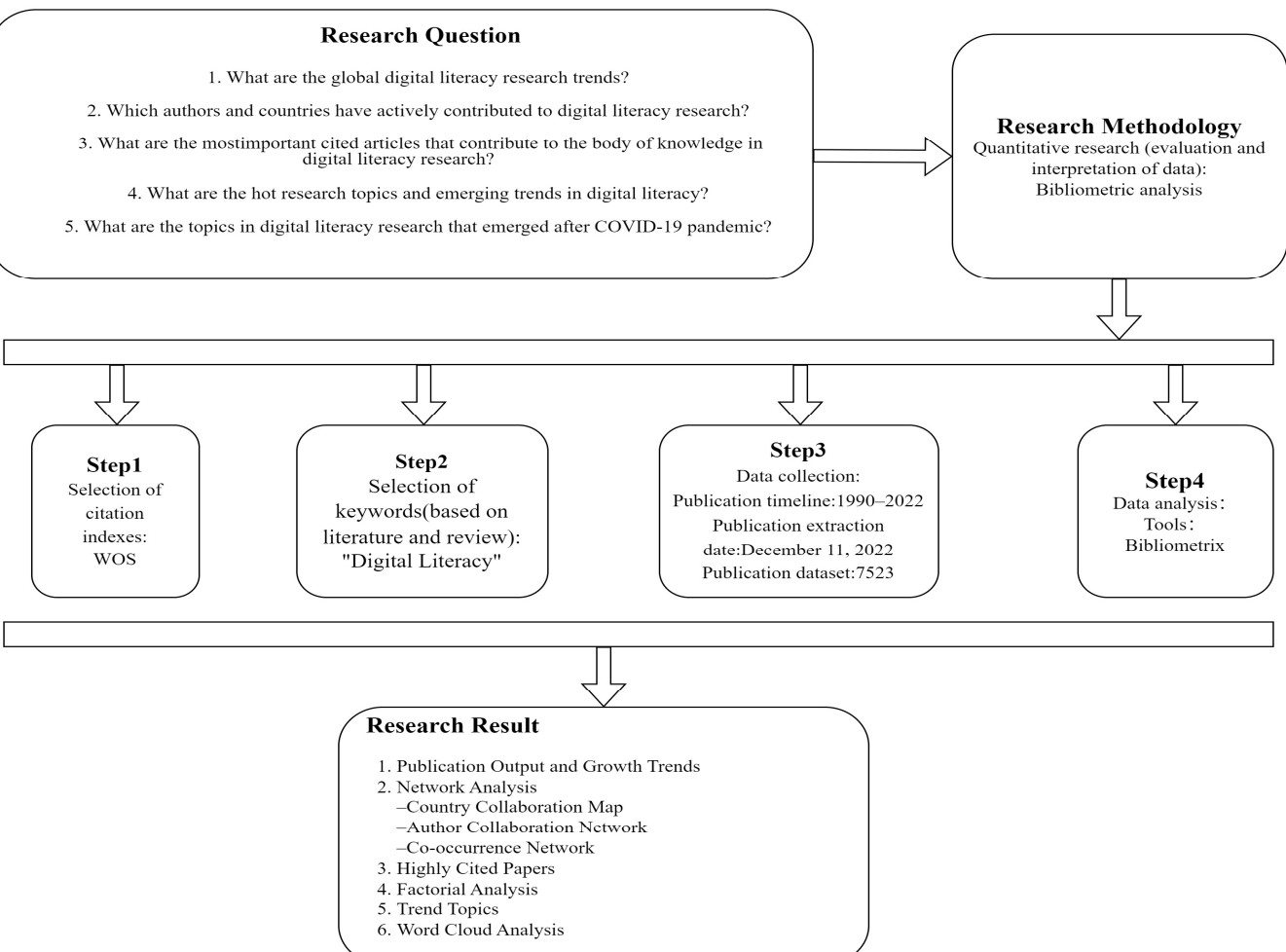

**Figure 1.** Stages of bibliometric analysis of research on digital literacy.

This study utilized Bibliometrix, which is a free software package developed by Massimo Aria and Corrado Cuccurullo [45,56]. Bibliometrix for RStudio is equipped with a built-in utility, Biblioshiny, which provides a graphical interface for non-coders to perform comprehensive analyses through improvised drawing representations. The software combines various bibliometric techniques, such as co-citation network analysis, generative collaboration networks, and factorial analysis, among others.

## 4. Results

### 4.1. Publication Output and Growth Trends

The annual publications depicted in Figure 2 offer valuable insights into the advancements achieved in digital literacy research. Since its inception in the early 1990s, the number of publications on this subject has increased annually, with a growth rate of 25.96%. Particularly in recent years, this growth has accelerated, resulting in a total of 1281 publications in 2022, as identified by our search. This number is expected to surpass the number of publications in 2021, which stood at 1285.

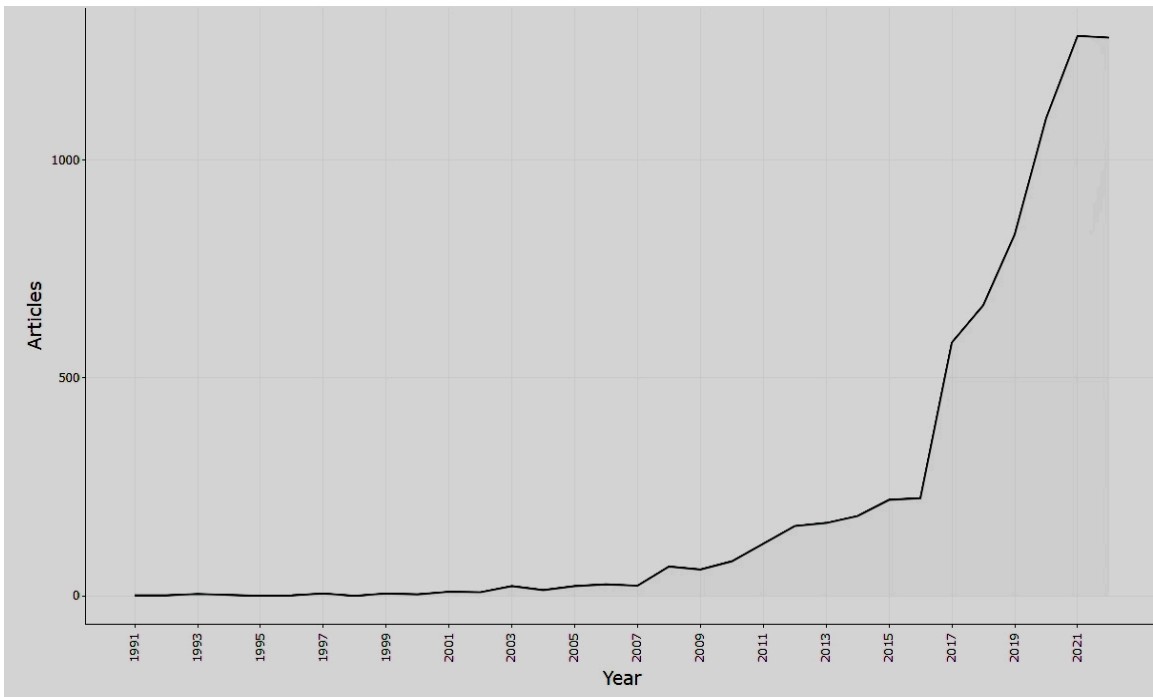

**Figure 2.** Annual scientific production.

To comprehend this trend, it is necessary to examine the leading sources, countries, affiliations, and keywords across all 7523 considered documents. Table 2 presents distinct analyses of the top 10 countries, references, authors, associations, and keywords.

**Table 2.** Listing of the top 10 contributing countries, sources, authors, affiliated institutions, and keywords.

| Countries | F | Sources | F | Authors | F | Affiliations | F | Keyword | F |
|---|---|---|---|---|---|---|---|---|---|
| USA | 4667 | *Journal Of Adolescent & Adult Literacy* | 187 | Kucirkova, N. | 21 | UNIV SYDNEY | 98 | Literacy | 1016 |
| Spain | 1369 | *Journal Of Medical Internet Research* | 144 | Okan, O. | 21 | UNIV TORONTO | 94 | Education | 538 |
| UK | 1263 | *Computers & Education* | 113 | Dadaczynski, K. | 20 | DEAKIN UNIV | 85 | Technology | 504 |
| Australia | 1224 | *International Journal of Environmental Research and Public Health* | 89 | Merchant, G. | 15 | UNIV MELBOURNE | 85 | Information | 458 |
| Canada | 913 | *Reading Teacher* | 89 | Tomczyk, L. | 15 | UNIV BRITISH COLUMBIA | 83 | Students | 401 |
| China | 886 | *Comunicar* | 84 | Aguaded, I. | 14 | UNIV CALIF SAN FRANCISCO | 73 | Internet | 360 |
| Germany | 750 | *Sustainability* | 79 | Mills, K.A. | 14 | UNIV OSLO | 73 | Skills | 302 |
| Brazil | 516 | *Education And Information Technologies* | 71 | Smith, B.E. | 14 | MONASH UNIV | 68 | Knowledge | 276 |
| Netherlands | 346 | *Nordic Journal of Digital Literacy* | 68 | Burnett, C. | 13 | UNIV MARYLAND | 63 | Impact | 245 |
| Portugal | 332 | *Learning Media and Technology* | 64 | Jiang, L.J. | 13 | NANYANG TECHNOL UNIV | 62 | Media | 206 |

The United States published the majority of articles on digital literacy (4667, n = 7523, accounting for 62.0%), followed by Spain (1369, n = 7523, accounting for 18.2%), and the United Kingdom (1263, n = 7523, accounting for 16.8%). Developed European countries

and the United States conducted most of the research on digital literacy. Among the 7523 sources considered, Wiley-Blackwell's *Journal of Adolescent and Adult Literacy*, edited by Syracuse University's Kathleen A. Hinchman and Kelly Chandler-Olcott, published the most papers, registering 187 (187, n = 216, or 2.5%). JMIR Publications' *Journal of Medical Internet Research* followed with 144 articles, and Elsevier's *Computers and Education* with 113 articles. The selected articles had a total of 19,761 authors, with Kucirkova, N. (21 pieces), Okan, O. (21 pieces), Dadaczynski, K. (20 pieces), Merchant, G. (15 articles), and Tomczyk, L. (15 papers) being the top authors (note: these authors may not necessarily have the most citations).

The top-ranking institutions in the literature were the University of Sydney (98), the University of Toronto (94), Deakin University (85), the University of Melbourne (85), and the University of British Columbia (83 articles). The authors used a total of 14,475 keywords, with the most frequently used being literacy (1016), education (538), technology (504), information (458), students (401), Internet (360), skills (302), knowledge (276), influence (245), and media (206). These findings suggest a clear research interest in digital literacy education among researchers.

*4.2. Network Analysis*

4.2.1. Country Collaboration Map

Figure 3 illustrates the collaborations among countries in digital literacy research. Virtually every region of the world now participates in the digital literacy research network, with denser network nodes in Europe and North America, indicating that these regions are at the center of the cooperation network, and that digital literacy researchers in these regions have closer collaborations. Additionally, China, Australia, and some countries in Africa and South America also play a role in the cooperation network.

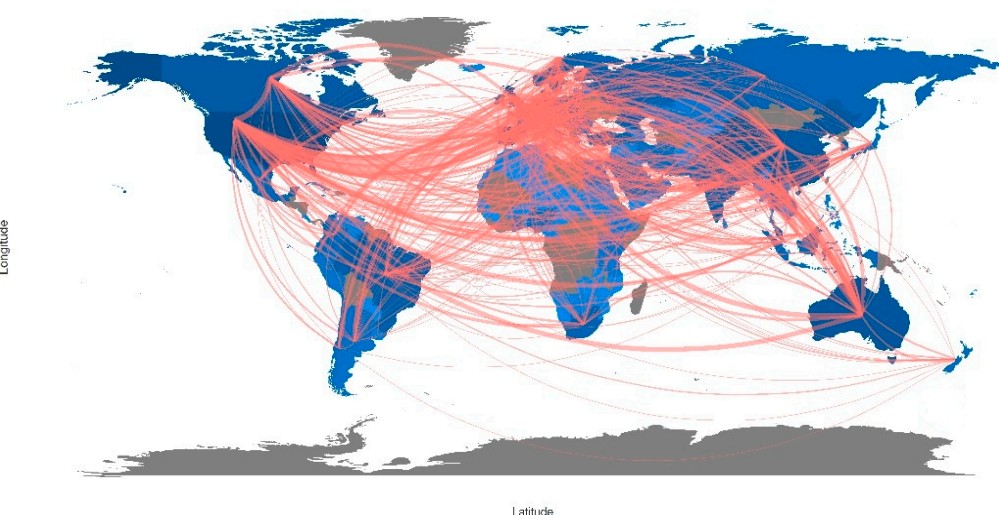

**Figure 3.** Country collaboration map.

4.2.2. Author Collaboration Network

An analysis of author collaborations is a useful indicator of a topic's research potential and was utilized to assess progress on the topic of digital literacy [53]. Figure 4 displays the authors with the closest research cooperation, with Okan, O. andDadaczynski, K. having the highest level of collaboration, followed by Van Deursen, A.J.A.M. andVan Dijk, J.A.G.M., and then Robinson, L. and Bawden, D. A significant positive correlation was found between the degree of cooperation among authors and the academic influence of their scientific research results [57]. Combined with the information on highly cited documents by authors in the next section, this suggests a specific relationship between the two.

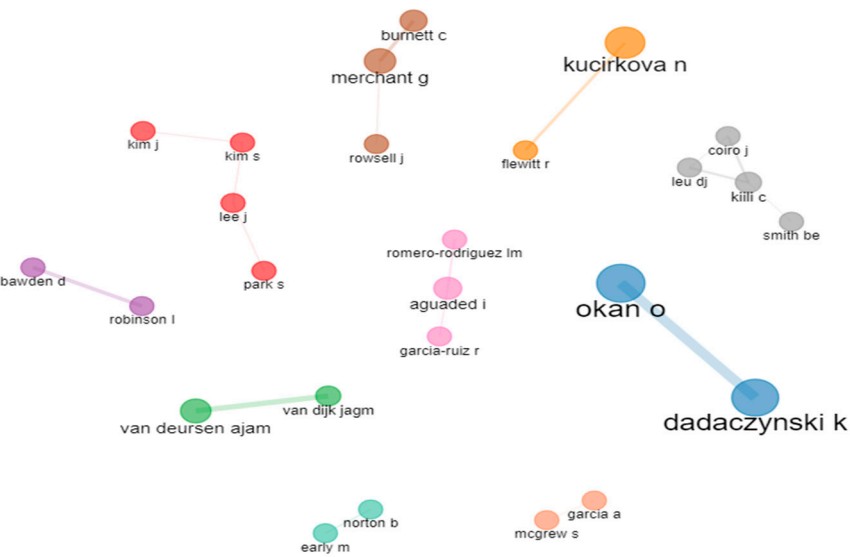

**Figure 4.** Collaboration network.

### 4.2.3. Co-Occurrence Network

A co-occurrence analysis of keywords is a useful tool for representing the domain space and primary content of a specific theme. Keywords Plus is an index automatically generated by WoS based on the titles of references in papers. This index requires the search term to appear multiple times in a bibliography and is sorted from phrases composed of numerous different words into a single search term. Keywords Plus can be considered an enhancement of traditional keyword or title searches.

We conducted a keyword co-occurrence analysis with Keywords Plus, and the results are shown in Figure 5. The keyword with the highest frequency was "literacy". Keywords were grouped into two distinct clusters. The first cluster included closely related keywords such as literacy, education, technology, information, students, and skills. The second cluster consisted of keywords such as Internet, care, communication, digital divide, access, and health. These two sets of keywords provide a comprehensive overview of the domain space and primary content of digital literacy research.

On the one hand, digital literacy research focuses on the digital divide resulting from the Internet and information and communication technology (ICT), as well as research on the use of network tools. Among these, research on students' digital literacy education and skill improvement is currently the most widely discussed research direction.

### 4.3. Highly Cited Papers

Highly cited papers are generally articles with high academic value and significant professional influence. As shown in Table 3, although review articles accounted for a small proportion of the total number of article reports (5.8%), three of them were highly cited papers (30%).

The articles included in this study predominantly comprised survey research and review articles. For example, Livingstone, S. et al. investigated the digital divide among children and young people by analyzing the national survey results of British teenagers aged between 9 and 19 years. They discovered that age, gender, and socioeconomic status inequalities had an impact on the quality of their access to and use of the Internet [58]. Similarly, Choi, N.G. et al. surveyed internet usage patterns, eHealth literacy, and attitudes towards computer/Internet use among low-income home-based older adults. The authors found that these individuals exhibited lower rates of internet use compared to the average, which was likely due to a lack of access to computer/Internet technology, a lack of financial resources for acquiring computers and technology, medical conditions limiting access such as a disability and associated pain, or a lack of knowledge and skills for using the Internet [59].

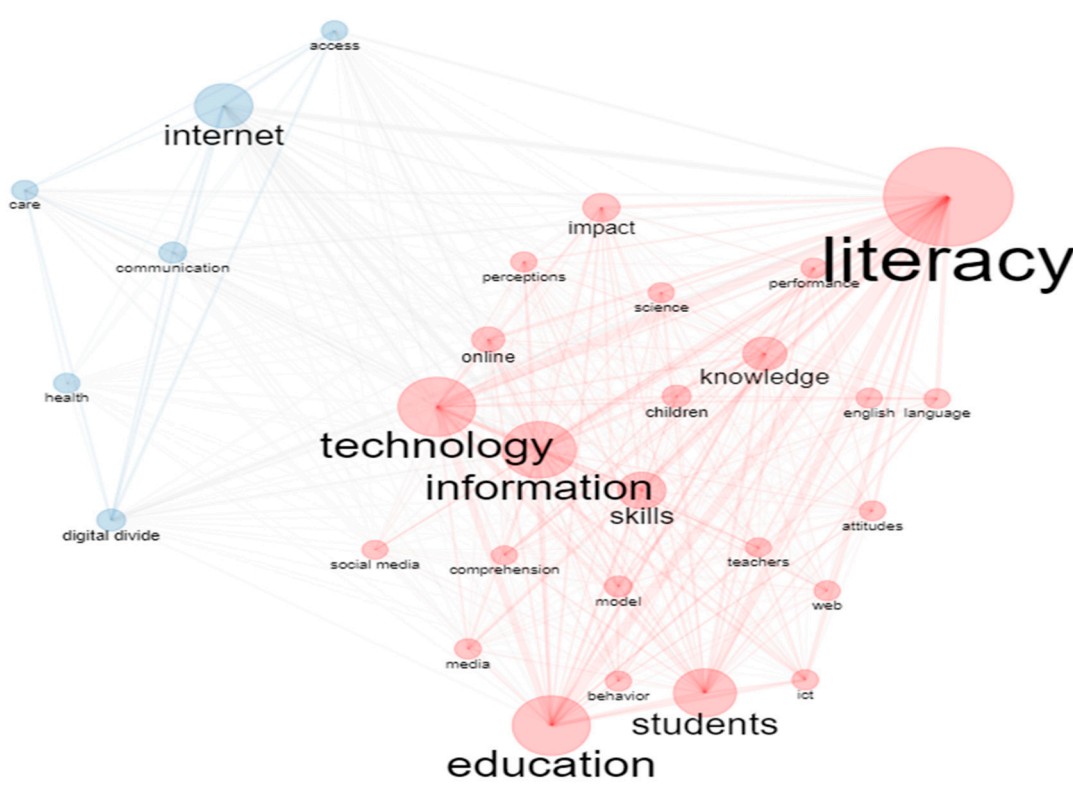

**Figure 5.** Co-occurrence network.

**Table 3.** Top 10 most highly cited papers.

| Title | Details | Total Cited | Annual Cited |
|---|---|---|---|
| Gradations in digital inclusion: children, young people, and the digital divide | LIVINGSTONE S, 2007, NEW MEDIA SOC | 584 | 36.5 |
| The dark side of information: overload, anxiety, and other paradoxes and pathologies | BAWDEN D, 2009, J INF SCI | 506 | 36.14 |
| The relation between 21st-century skills and digital skills: A systematic literature review | VAN LAAR E, 2017, COMPUT HUM BEHAV | 422 | 70.33 |
| Internet skills and the digital divide | VAN DEURSEN AJAM, 2011, NEW MEDIA SOC | 402 | 33.5 |
| Information and digital literacies: a review of concepts | BAWDEN D, 2001, J DOC | 389 | 17.68 |
| Identity, language learning, and social change | NORTON B, 2011, LANG TEACHING | 375 | 31.25 |
| Children and Adolescents and Digital Media | CHASSIAKOS YR, 2016, PEDIATRICS | 359 | 51.29 |
| eHealth Literacy: Extending the Digital Divide to the Realm of Health Information | NETER E, 2012, J MED INTERNET RES | 356 | 32.36 |
| The Digital Divide Among Low-Income Homebound, Older Adults: Internet Use Patterns, eHealth Literacy, and Attitudes Toward Computer/Internet Use | CHOI NG, 2013, J MED INTERNET RES-a | 347 | 34.7 |
| eHealth Literacy and Web 2.0 Health Information Seeking Behaviors Among Baby Boomers and Older Adults | TENNANT B, 2015, J MED INTERNET RES | 322 | 40.25 |

In another study, Van Deursen, A.J.A.M. investigated the increasing importance of internet operations and skills in the contemporary information society, and how they increasingly determine people's status in the labor market and social life. The study revealed that people with lower education levels are quickly excluded from the information society [60].

Neter, E. et al. documented differences in contextual attributes, information consumption, and information search results between respondents with high and low levels of

eHealth literacy [61]. The authors suggest that the association of eHealth literacy with contextual attributes and characteristics reinforces existing social differences. More comprehensive education of at-risk groups and those in need, such as people with chronic diseases, is necessary. Additionally, technology should be designed in a format that suits more consumers.

Tennant, B. et al. found that younger and more educated individuals had higher eHealth literacy among baby boomers and older adults. Women and those with higher education, particularly at the graduate level, were more likely to use Web 2.0 for health information [62]. The authors also discussed the concepts of "information literacy" and "digital literacy" [2], as well as the dark side of information, including overload, anxiety, and other paradoxes and pathologies [63].

Van Laar, E et al. conducted a systematic literature review on the relationship between 21st century skills and digital skills. The authors identified seven core skills: technology, information management, communication, collaboration, creativity, critical thinking, and problem-solving. They also identified five contextual skills: ethical awareness, cultural awareness, flexibility, self-direction, and lifelong learning [64].

Overall, the articles included in this study shed light on the digital divide, eHealth literacy, and the importance of technology and digital skills in the information society. They also highlight the need for comprehensive education and technology design that suits all consumers.

These highly cited articles are not only review articles of literature but also investigations and applications. These review articles provide detailed analysis of digital literacy, offering a comprehensive understanding of the concept of digital literacy, its development, and implications. Moreover, they provide important insights into the current trends and future development of digital literacy. In addition, highly cited articles also include application articles, which demonstrate the application of digital literacy in different areas. These articles help to understand how digital literacy can be applied in various fields, such as education, health, and business. Furthermore, these articles provide useful guidelines for the development of digital literacy in various areas.

### 4.4. Factorial Analysis

In this study, we utilized factorial methods, specifically multiple correspondence analysis, to examine the Keywords Plus for digital literacy research. The results, illustrated in Figure 6, revealed that digital literacy research could be categorized into two distinct themes: Dimension 1 (Dim1 = 38.85%) and Dimension 2 (Dim2 = 21.74%). These two dimensions accounted for a significant proportion of the analyzed data, indicating their importance in defining the data. In fact, the combination of these two dimensions explained 60.59% of the analyzed data. Thus, it can be concluded that these two dimensions played a crucial role in shaping the landscape of digital literacy research.

The factorial analysis of Keywords Plus reveals two dimensions of digital literacy research. However, clarifying the sub-themes and connections between them is challenging. To obtain a more intuitive understanding, we converted the conceptual structure diagrams from the multiple correspondence analysis into a thematic tree diagram for analysis, as illustrated in Figure 7. By examining the branch relationships in the dendrogram, we identified the following themes: (1) ICT (information and communications technology), computer self-efficacy, and gender gap topics; (2) the digital divide in Internet use (Internet, web, online information, digital divide, access, and internet use); (3) health literacy and health management topics (outcomes, healthcare literacy, communication and quality, health and management); (4) social media usage behaviors of children and minors (interventions, impact and behavior, social media and support, engagement, literacy and adolescents, media and children); (5) digital literacy in English learning (literacies and school, language and English); (6) student classroom education (classroom, students, comprehension, science and instruction, design, framework, education and knowledge); and (7) digital literacy for teachers (perceptions and performance, skills and attitudes, technology and model,

information literacy, beliefs, teacher and competence). Figure 7 shows that themes (3) and (4) have a strong correlation, while articles (6) and (7) are closely related.

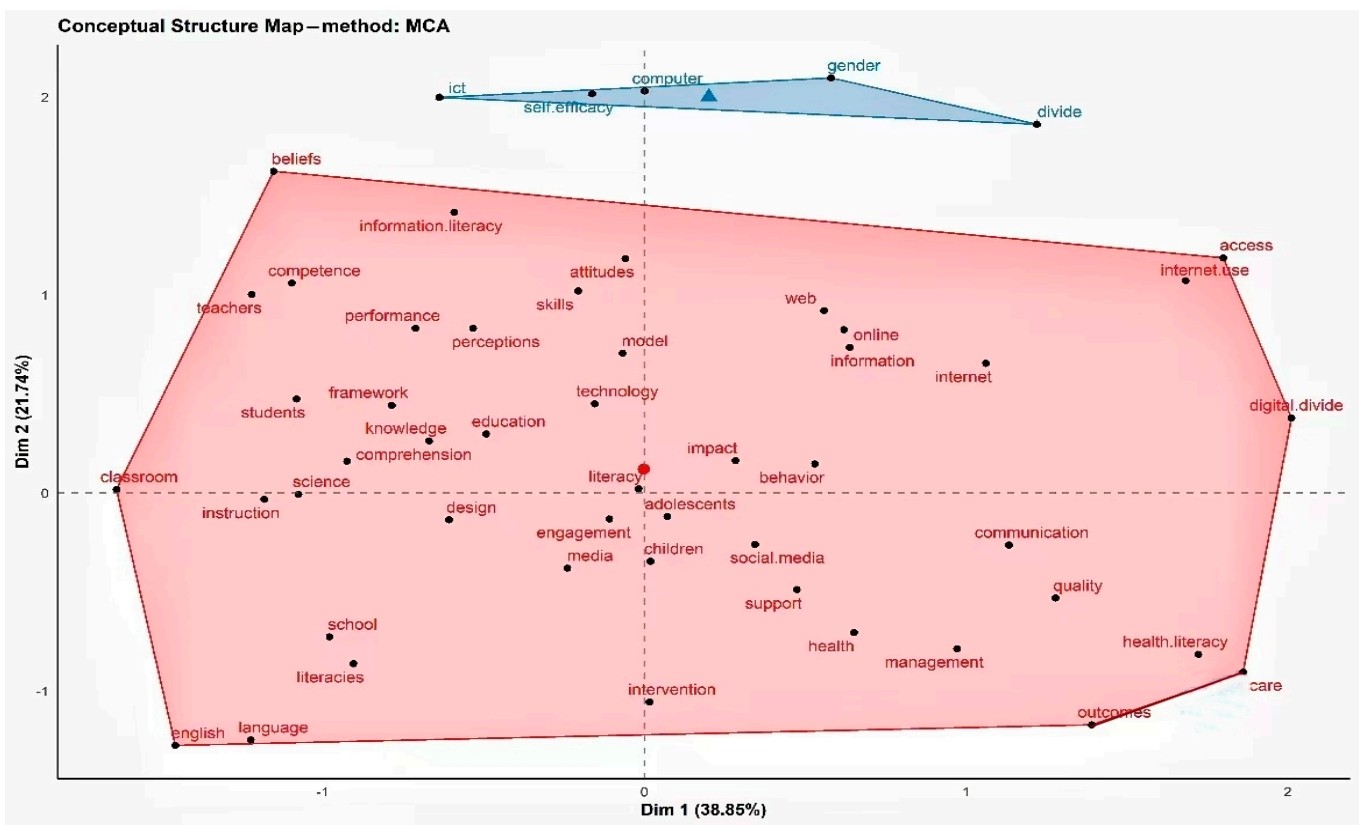

**Figure 6.** Multiple correspondence analysis of Keywords Plus.

*4.5. Trend Topics*

In the information age, the progress of digital literacy research is intricately linked to the continuous advancements in Internet technology and information and communication technologies (ICT). While these technological developments have brought convenience to society, they have also generated significant concerns. Consequently, these issues have become a central area of focus within digital literacy research.

Figure 8 illustrates that between 2003 and 2008, the primary research focus within digital literacy was information search behavior, specifically people's abilities to search for and retrieve information. By 2006, research enthusiasm in this area had reached its peak. Since 2012, however, digital literacy research has diversified significantly, covering topics such as democracy, university libraries, blogs, numeracy, and computer use. Emerging issues have quickly garnered attention, resulting in digital divide issues, including web and gender, becoming recent research hotspots. To tackle these issues, digital literacy, skills, education, and training have increasingly become areas of focus for research. Furthermore, the outbreak of COVID-19 has also resulted in telemedicine and COVID-19 becoming research hotspots over the past two years. Finally, it is worth noting that issues and scientific literacy faced by individuals in the information society have remained long-term research hotspots within digital literacy.

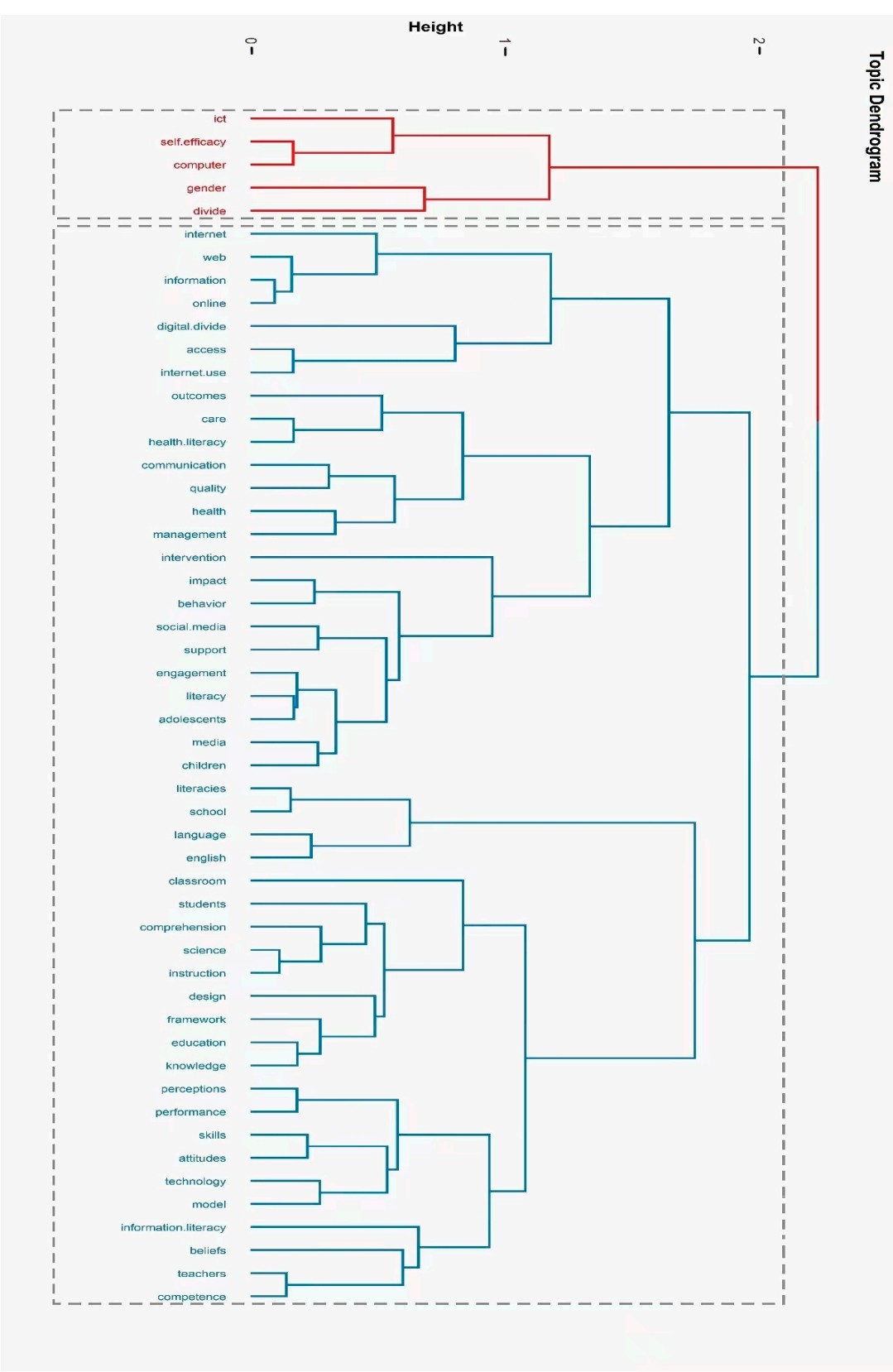

**Figure 7.** Dendrogram of digital literacy research topics.

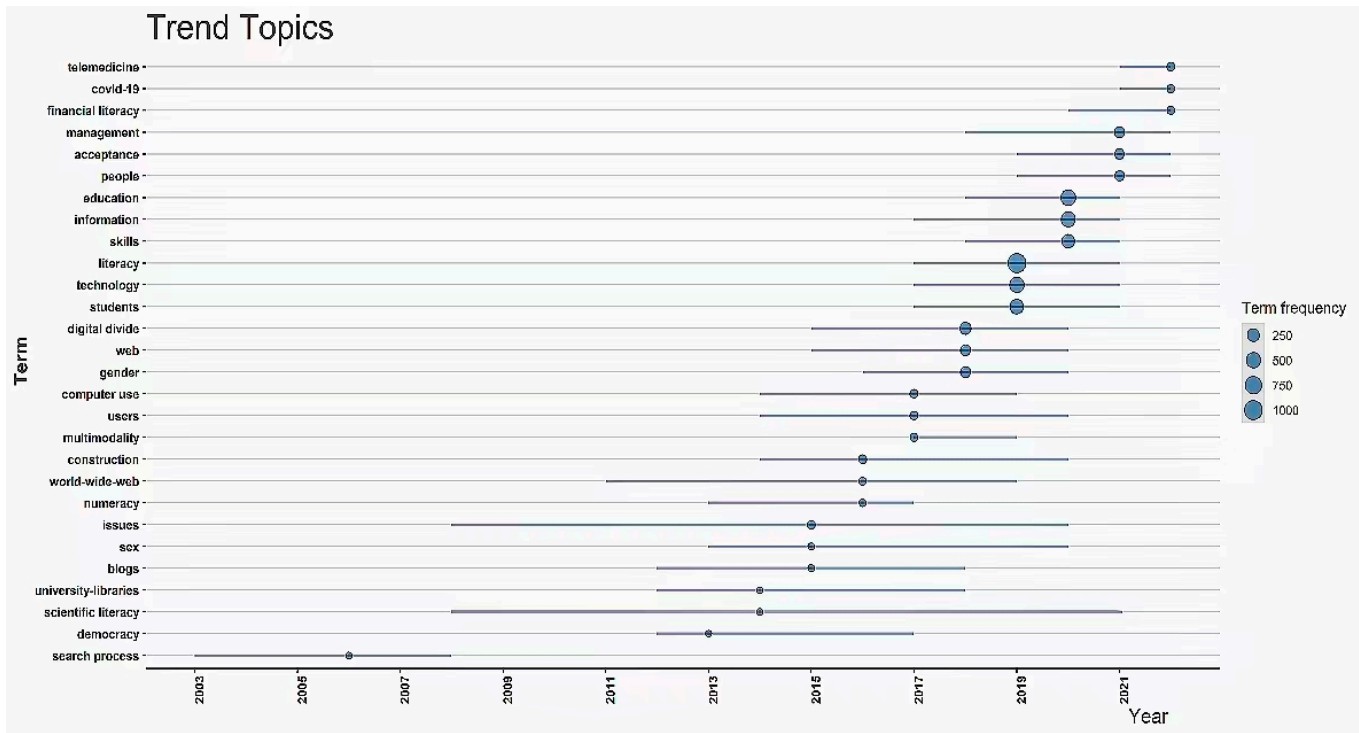

**Figure 8.** Trends in digital literacy topics.

*4.6. Word Cloud Analysis*

To investigate the evolving themes in digital literacy research during the COVID-19 pandemic, we analyzed 3961 articles published between 1 January 2020 and 31 December 2022, selecting papers related to the COVID-19 period from the same dataset. We analyzed the keywords used by the authors and identified the keywords that registered the highest frequency, as presented in Table 4. Our analysis revealed that "COVID-19" (273) and "health literacy" (153) emerged as high-frequency keywords in digital literacy research during the pandemic.

**Table 4.** Most frequent words.

| Words | Occurrences |
|---|---|
| digital literacy | 558 |
| COVID-19 | 273 |
| information literacy | 186 |
| media literacy | 159 |
| health literacy | 153 |
| higher education | 139 |
| social media | 133 |
| literacy | 126 |
| digital divide | 121 |
| education | 119 |

To identify emerging themes in digital literacy research, we conducted word cloud processing on the keywords. As depicted in Figure 9, emerging themes include "ehealth", "mhealth", "health literacy", "digital health literacy", "digital health", "fake news", and "online learning".

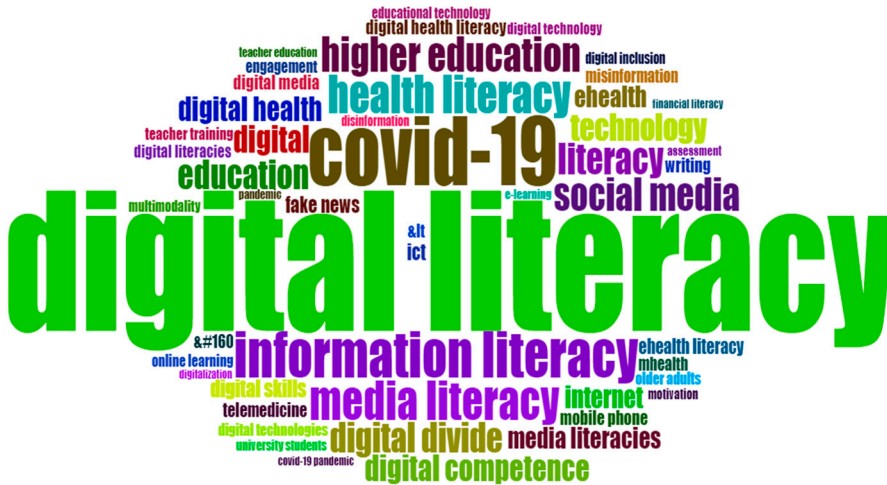

**Figure 9.** Word cloud.

## 5. Findings

In conclusion, this study provides a comprehensive overview of the current state of digital literacy research, identifying key trends, prominent publications and institutions, and emerging themes in the field. Our findings demonstrate the growing importance of digital literacy in today's society, especially given the rapid development of social media technologies and the recent shift to remote healthcare and online education due to the COVID-19 pandemic. The United States, Spain, and the United Kingdom are the leading countries in digital literacy research, with the *Journal of Adolescent and Adult Literacy*, the *Journal of Medical Internet Research*, and *Computers and Education* being the most influential publications in this area. The University of Sydney, the University of Toronto, Deakin University, the University of Melbourne, and the University of British Columbia are the most productive institutions in terms of digital literacy research publications. European and American countries exhibit a higher output of papers and pay more attention to digital literacy.

The Country Collaboration Map indicates that European and American countries have a greater degree of cooperation in the area of digital literacy research. The Author Collaboration Network shows that O. Okan and K. Dadaczynski have the most significant research collaboration, followed by A.J.A.M. Van Deursen and J.A.G.M. Van Dijk, and L. Robinson and D. Bawden. The degree of collaboration among authors positively correlates with the academic influence of their research output, as exemplified by articles authored by D. Bawden, A.J.A.M. Van Deursen, and J.A.G.M. Van Dijk. These articles provide a comprehensive analysis of digital literacy, including its development, implications, current trends, and future development, and demonstrate the application of digital literacy in various fields, such as education, health, and business.

The results of co-occurrence network and factor analysis suggest that digital literacy research can be divided into two dimensions: digital literacy issues and digital literacy education, with the possibility of further subdivisions as shown in the tree diagram. Additionally, there is a strong correlation between certain topics, such as topics 3 and 4, and topics 6 and 7. Digital literacy research can explore the integration of health literacy and social media, and research on digital literacy between teachers and students is inseparable.

COVID-19 and telemedicine have become prominent research topics in the past two years, and digital literacy research has adapted accordingly. The analysis of author keywords from 1 January 2020 to 31 December 2022 reveals that "COVID-19" and "health literacy" were the most frequent terms. Emerging themes include "ehealth", "mhealth", "digital health literacy", "digital health", "fake news", and "online learning". Remote healthcare, digital health, online education, and virtual classrooms are likely to remain research hotspots, and the issues arising from long-term remote working and teaching should be a new research area for digital literacy.

Overall, this study highlights the importance of digital literacy research in understanding and addressing the challenges and opportunities posed by the digital age. Bibliometric analysis is a valuable tool for identifying research trends and gaps, providing insights into the most influential publications, institutions, and authors. As the field of digital literacy research continues to evolve, it is essential to explore emerging themes and topics to stay relevant and innovative.

## 6. Discussion

Our study contributes significantly to the current understanding of digital literacy. Firstly, it provides a comprehensive overview of research trends and publication output on digital literacy, which can be used as a reference by researchers and practitioners interested in the field. Our analysis identifies the most prominent publications, institutions, and countries involved in digital literacy research, facilitating the identification of potential collaborators and research opportunities.

Secondly, the co-occurrence network and factor analysis used in this study offer insights into the different dimensions of digital literacy research. By identifying the various themes and sub-themes within digital literacy research, our study provides a conceptual framework that can guide future research. Researchers can utilize this framework to better comprehend the different aspects of digital literacy and focus their research accordingly.

Thirdly, our study identifies emerging themes in digital literacy research, including remote healthcare, digital health, online education, and virtual classrooms, which have become increasingly relevant due to the COVID-19 pandemic. By highlighting these emerging themes, our study provides directions for future research in digital literacy, which can help address the challenges and opportunities arising from the pandemic and other related trends.

Lastly, we demonstrate the potential of bibliometric analysis as a tool for understanding research trends and identifying research gaps in digital literacy. As digital literacy continues to evolve rapidly, bibliometric analysis can serve as a valuable tool for tracking these changes and guiding future research.

In summary, our study contributes significantly to the current understanding of digital literacy by providing a comprehensive overview of research trends and publication output, identifying different dimensions and emerging themes, and demonstrating the potential of bibliometric analysis as a research tool. This study offers valuable insights for researchers and practitioners interested in digital literacy and can guide future research efforts in this field.

## 7. Conclusions

To conclude, this study offers a comprehensive overview of the current state of digital literacy research, identifying key trends, prominent publications and institutions, and emerging themes in the field. Our findings underscore the significance of digital literacy in contemporary society, particularly in light of the COVID-19 pandemic and the shift towards remote healthcare and online education. Additionally, this study highlights the potential of bibliometric analysis as a valuable tool for comprehending research trends and identifying research gaps in digital literacy.

In the future, we intend to refine our research points and expand our data collection methods to further advance knowledge in this critical area of research. Our study provides a foundation for future research and offers valuable insights for researchers and practitioners who seek to deepen their understanding of digital literacy and its implications for society.

**Author Contributions:** Conceptualization, C.W. and L.S.; Methodology, C.W. and L.S.; Software, C.W.; Validation, C.W.; Formal analysis, C.W.; Investigation, C.W.; Resources, C.W.; Data curation, C.W.; Writing—original draft, C.W.; Supervision, L.S.; Project administration, L.S. All authors have read and agreed to the published version of the manuscript.

**Funding:** This research received no external funding.

**Institutional Review Board Statement:** Not applicable.

**Informed Consent Statement:** Not applicable.

**Data Availability Statement:** The data used in this research can be obtained from the corresponding authors upon request.

**Conflicts of Interest:** The authors declare no conflict of interest.

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
