# Peer review of "A Bibliometric Analysis of Digital Literacy Research from 1990 to 2022 and Research on Emerging Themes during the COVID-19 Pandemic"

_sustainability, doi:10.3390/su15075769_

Round 1
Reviewer 1 Report
I appreciate the authors’ efforts in the work put into the research of Analysis of Digital Literacy Research and Emerging Themes During COVID-19 Pandemic:
To enhance the quality of the study, the authors must do some revision of their research and pay attention to several important issues:
- The introduction should clearly state the research question(s) or the research objective(s) that the authors present for this paper.
- The paper should match the format of the journal.
- The research presented in the study has been carried out using Web of Science database. The authors should justify why was this used, instead of other recognised by scholars. In this direction, authors are advised to study and cite some updated works such as the following:
- Bibliometric Analysis of the Green Deal Policies in the Food Chain. Amfiteatru Econ. 2022, 24, 410–428. DOI:10.24818/EA/2022/60/410.
- Mapping Knowledge Area Analysis in E-Learning Systems Based on Cloud Computing. Electronics 2023, 12, 62. https://doi.org/10.3390/electronics12010062.
- Exploring the Research Regarding Frugal Innovation and Business Sustainability through Bibliometric Analysis. Sustainability. 2022, 14(3), 1326. https://doi.org/10.3390/su14031326.
- I recommend a more complex section of Conclusions, where authors could detail the information. Also, there are many paragraphs formed of single sentences. Paragraphs should include at least two sentences or phrases that describe the same idea.
- Authors need to explain about limitations and future research directions as well in revised draft.
Author Response
Response to Reviewer 1 Comments
Point 1: The introduction should clearly state the research question(s) or the research objective(s) that the authors present for this paper.
Response 1: Thanks for your suggestion. We have rewritten the Introduction according to the content of the literature and the standard format of the journal: The research seeks to answer the following six research questions:
RQ1. What are the global digital literacy research trends?
The answer to this question will help researchers investigate the development trends and status quo of digital literacy research.
RQ2. Which authors and countries have contributed actively to digital literacy research?
The answer to this question will empower scholars to recognize collaborations worldwide and potential collaborators on digital literacy research.
RQ3. What are the most important cited articles that add to the body of knowledge in the field of digital literacy research?
The answer to this question will enable researchers to choose an appropriate reference as a basis for their research.
RQ4. What are the hot research topics and emerging trends discussed in digital literacy?
The answer to this question would facilitate scholars to explore new research domains, research gaps, and potential topics in digital literacy.
RQ5. What are the topics in digital literacy research that emerged after the COVID-19 pandemic?
The answer to this question will help scholars to focus on the topics that can guide the academicians, practitioners, and managers to integrate digital literacy with education, healthcare, and governance during the COVID-19 pandemic.
RQ6. What new topics emerged in digital literacy research over the years?
The answer to this question would help budding researchers to recognize significant developments that have happened over the past year and new topics that have emerged in digital literacy research.
Point 2: The paper should match the format of the journal.
Response 2: According to your suggestions ,we revised and checked the full text according to the journal sample format.
Point 3: The research presented in the study has been carried out using Web of Science database. The authors should justify why was this used, instead of other recognised by scholars.
Response 3: The literature you recommended is of great reference value, and we have added it to the bibliography. In addition, we also referred to the product introduction webpage of the WOS core database to prove that the WOS database can be used for research. At the end of the article, we also wrote that the scope of data search would be further expanded, not just the WOS database.
Point 4: I recommend a more complex section of Conclusions, where authors could detail the information. Also, there are many paragraphs formed of single sentences. Paragraphs should include at least two sentences or phrases that describe the same idea..
Response 4: According to your suggestions, we have expanded and revised the content of the conclusion section and the single-sentence paragraphs. We have also used the English editing service recommended by the journal to improve the English writing level of the full text.
Point 5: Authors need to explain about limitations and future research directions as well in revised draft.
Response 5: According to your suggestions,we added and revised the limitations of this paper and proposed future research directions at the end of the article.
Reviewer 2 Report
Authors propose a bibliometric analysis of digital literacy in order to establish patterns related with trends in research topics. The objective of manuscript could be a relevant contribution for future research.
In the methodology is not clear because authors has been analyzed papers since 1990, which event motivates this year selection. They also consider the years 2019 to 2022 in the study, but they are atypical years and they should be treated separately.
Authors also focus the analysis over WoS, but why they consider that only data source and not other scientific sources is not clear in their methodology.
The sctientific procedure for establishing the literature review process is not clear in the methodology.
In the discussions, the contribution is not appreciated, authors should show why the relevant topics in digital literacy were selected for researches.
Figures 6, 7 and 8 are blurry and difficult to read. Figure 9 does not contribute to the analysis. Also Figure 9 is produced by mixing years with different behaviors, the world cloud have words like covid-19 that were not relevant prior to the year 2018.
The discussion should be strengthened to define how the results can be the basis for future researches.
Author Response
Response to Reviewer 2 Comments
Point 1: Authors propose a bibliometric analysis of digital literacy in order to establish patterns related with trends in research topics. The objective of manuscript could be a relevant contribution for future research.
Response 1: Thanks for your suggestions, we have revised and rewritten the Abstract and introduction sections according to the journal format and article content. Part of the content is as follows, please refer to the revised draft for specific revisions:
Abstract:Background: Digital literacy is an increasingly important research topic today. Digital advances are supporting and accelerating the achievement of sustainable development, while at the same time, inequalities resulting from uneven diffusion of new technologies are emerging, especially as the world faces one of the greatest public health crises of the century, the COVID-19 pandemic. The primary purpose of this paper is to conduct an in-depth analysis of scientific production using bibliometrics, to understand the current research status of studies, and evaluate the pandemic's impact on digital literacy research. Methods: 7523 documents published between 1990 and 2022 were identified in the Web of Science database. This literature was analyzed using bibliometric research methods such as growth trend analysis, network analysis, highly cited literature analysis, factor analysis, and time series-based hot topic analysis. Additionally, a word cloud analysis of the keywords of digital literacy literature from 2020-2022 during the COVID-19 pandemic was performed. Results and Conclusions: Collaborative networks, key themes, and research trends in digital literacy research were identified, and the impact of the pandemic on digital literacy research was assessed.
Point 2: In the methodology is not clear because authors has been analyzed papers since 1990, which event motivates this year selection. They also consider the years 2019 to 2022 in the study, but they are atypical years and they should be treated separately.
Response 2: Thank you so much for your suggestion; we have added and stressed the selection of time nodes in the article. The year 1990 was chosen as the node because it was found through the previous literature review that the discussion of the concept of digital literacy research only began to appear in the 1990s. Generally speaking, the start time of the new crown was between October 6, 2019, and December 11, 2019, so we chose January 2020 to December 2022 as the starting and ending points for selecting literature data during the COVID-19 period.
Point 3: Authors also focus the analysis over WoS, but why they consider that only data source and not other scientific sources is not clear in their methodology.
Response 3:. According to your suggestions ,we've added some references to support it. In addition, we also refer to the product introduction webpage of the WOS core database to prove that the WOS database can be used for research. At the end of the article, we also wrote that the scope of data search will be further expanded in future research, not just the WOS database.
Point 4: The sctientific procedure for establishing the literature review process is not clear in the methodology..
Response 4: In order to have a clearer literature review, we introduce a table to present the literature review
Point 5: Figures 6, 7 and 8 are blurry and difficult to read. Figure 9 does not contribute to the analysis. Also Figure 9 is produced by mixing years with different behaviors, the world cloud have words like covid-19 that were not relevant prior to the year 2018..
.
Response 5: We have improved the clarity of Figures 6, 7, and 8 using software . The data source in Figure 9 is nearly three years after the outbreak of the new crown (from January 1, 2020, to December 31, 2022). To further highlight this part of the content, we have made the following revisions "To more clearly reveal emerging themes in digital literacy research In the context of the COVID-19 pandemic, we reviewed literature data from 3961 articles from January 1, 2020, to December 31, 2022.”
Point 6: The discussion should be strengthened to define how the results can be the basis for future researches.
.
Response 6: Thanks for your suggestions; we have made further improvements to the conclusions.
Reviewer 3 Report
I read this paper a few times and still have problems catching the idea for what purpose it was written. The question “So what?” given to the Authors of every academic paper, remains unanswered. Therefore the main problem with this paper is that it does not deliver a scientific contribution to the field of digital literacy and to any other field unfortunately.
The disputable value of the paper starts with the title. It suggests that we will find out what were the emerging themes in digital literacy research during the COVID-19 pandemic. Unfortunately, the Authors analyzed 7523 articles published since the 1990s, so the temporal scope of the analyzes is different. According to the title, the analyzed papers should be dated starting from the beginning of the COVID-19 pandemic (the end of 2019/beginning of 2020). But the content related to this period has not appeared until lines 267-269. And it ended in line 282. I find it insufficient because it does not deliver anything new actually.
My detailed comments are below:
· Purpose – “This paper conducts a bibliometric analysis of digital literacy” – paper is not a human or computer so it cannot conduct any analysis. It can present, deliver, etc. However, the objectives of scientific papers should be more sophisticated.
· Abstract is really hard to comprehend (especially lines 14-19). I do not understand, what the Authors meant.
· Lines 27-28 – Authors deliver numbers, without giving any references.
· Lines 36-37 – “The main purpose of this study is to help researchers sort out the development of digital literacy research and discover new themes that have emerged in digital literacy research in the context of the pandemic” – we have a different objective here and it is totally different than stated in the abstract. Moreover, this objective is not reached in the paper.
· Lines 37-39 – “The research will also help coordinate research networks across countries, authors, and universities. This paper attempts to achieve this research goal by using related techniques of bibliometrics” – I am wondering, how do you coordinate networks of researchers by bibliometrics techniques?
· Section 2.2 – The content should be better presented as it is difficult to find the most important constructs or research results in the field of digital literacy. Maybe a table would be more appropriate to show the results of your analyses.
· Lines 202-228, 238-248 – Too much information given in a way that is hard to follow. It would help if you worked on the presentation of your results, as it is challenging to stay focused while reading it.
· Line 252 – “The research results of researchers are often closely related to the development of real society” – I do not understand the usage of the term “the real society” here. Plus it is quite obvious that if we deal with the area of social science the results will be related to the changes in society’s functioning.
· 5. Discussion and Conclusions – Again, poorly presented. It is not an academic standard of writing.
· References – Impropriate format while citing within the text.
Summing up, I do not find a scientific contribution in this work and I do not think that the text is in accordance with academic writing standards.
Author Response
Response to Reviewer 3 Comments
Point overall
Authors propose a bibliometric analysis of digital literacy in order to establish patterns related with trends in research topics. The objective of manuscript could be a relevant contribution for future research.
Response overall:
First of all, thank you for your suggestion. The title is indeed ambiguous. We have carefully considered the title and sought help from English editing services to try to eliminate the ambiguity so that the title can match the article's content.
Your suggestions are specific and effective and have obvious help in improving the level of the article. The specific questions and answers are as follows:
Point 1:
Purpose – “This paper conducts a bibliometric analysis of digital literacy” – paper is not a human or computer so it cannot conduct any analysis. It can present, deliver, etc. However, the objectives of scientific papers should be more sophisticated.
- Abstract is really hard to comprehend (especially lines 14-19). I do not understand, what the Authors meant.
Response 1: Thank you for your suggestion. We have rewritten and revised the abstract according to the article content and journal format:
Abstract:Background: Digital literacy is an increasingly important research topic today. Digital advances are supporting and accelerating the achievement of sustainable development, while at the same time, inequalities resulting from uneven diffusion of new technologies are emerging, especially as the world faces one of the greatest public health crises of the century, the COVID-19 pandemic. The primary purpose of this paper is to conduct an in-depth analysis of scientific production using bibliometrics, to understand the current research status of studies, and evaluate the pandemic's impact on digital literacy research. Methods: 7523 documents published between 1990 and 2022 were identified in the Web of Science database. This literature was analyzed using bibliometric research methods such as growth trend analysis, network analysis, highly cited literature analysis, factor analysis, and time series-based hot topic analysis. Additionally, a word cloud analysis of the keywords of digital literacy literature from 2020-2022 during the COVID-19 pandemic was performed. Results and Conclusions: Collaborative networks, key themes, and research trends in digital literacy research were identified, and the impact of the pandemic on digital literacy research was assessed.
Point 2: Lines 36-37 – “The main purpose of this study is to help researchers sort out the development of digital literacy research and discover new themes that have emerged in digital literacy research in the context of the pandemic” – we have a different objective here and it is totally different than stated in the abstract. Moreover, this objective is not reached in the paper.
- Lines 37-39 – “The research will also help coordinate research networks across countries, authors, and universities. This paper attempts to achieve this research goal by using related techniques of bibliometrics”– I am wondering, how do you coordinate networks of researchers by bibliometrics techniques?
.
Response 2: Thank you for your suggestion. We have rewritten and revised the Introduction according to the article content and journal format. For details, please refer to the revised manuscript.
The research seeks to answer the following six research questions:
RQ1. What are the global digital literacy research trends?
The answer to this question will help researchers investigate the development trends and status quo of digital literacy re-search.
RQ2. Which authors and countries have contributed actively to digital literacy research?
The answer to this question will empower scholars to recognize collaborations worldwide and potential collaborators on digital literacy research.
RQ3. What are the most important cited articles that add to the body of knowledge in the field of digital literacy research?
The answer to this question will enable researchers to choose an appropriate reference as a basis for their research.
RQ4. What are the hot research topics and emerging trends discussed in digital literacy?
The answer to this question would facilitate scholars to explore new research domains, research gaps, and potential topics in digital literacy.
RQ5. What are the topics in digital literacy research that emerged after the COVID-19 pandemic?
The answer to this question will help scholars to focus on the topics that can guide the academicians, practitioners, and managers to integrate digital literacy with education, healthcare, and governance during the COVID-19 pandemic.
RQ6. What new topics emerged in digital literacy research over the years?
The answer to this question would help budding researchers to recognize significant developments that have happened over the past year and new topics that have emerged in digital literacy research.
Point 3: Section 2.2 – The content should be better presented as it is difficult to find the most important constructs or research results in the field of digital literacy. Maybe a table would be more appropriate to show the results of your analyses.
Response 3:
According to your suggestion,in order to have a clearer literature review, we introduce a table to present the literature review
Point 4: Lines 202-228, 238-248 – Too much information given in a way that is hard to follow. It would help if you worked on the presentation of your results, as it is challenging to stay focused while reading it.
.
Response 4: According to your suggestions, we have sought English-language editing services to revise the text of this article.
Point 5: Line 252 – “The research results of researchers are often closely related to the development of real society” – I do not understand the usage of the term “the real society” here. Plus it is quite obvious that if we deal with the area of social science the results will be related to the changes in society’s functioning.
Response 5: I'm sorry for such a mistake in the article.We have modified it as follows.
“We are in the Information Age. The development of digital literacy research is closely related to the development of Internet technology and ICT. The rapid growth of Internet technology and ICT has brought convenience to people but also some problems. These problems have become a focus of digital literacy research.”
Point 6: Discussion and Conclusions – Again, poorly presented. It is not an academic standard of writing..
Response 6: According to your suggestions, we have re-added and revised the Discussion and Conclusions sections.
Point 7: References – Impropriate format while citing within the text..
Response 7: According to your suggestions, we revised the references based on the journal sample.
Reviewer 4 Report
The current research paper titled “A Bibliometric Analysis of Digital Literacy Research and Emerging Themes During COVID-19 Pandemic”, argued that this paper conducts a bibliometric analysis of digital literacy. It mainly analyzes digital literacy research publishing trends, collaborative networks, highly cited literature, main research themes, hot topics, and emerging themes. A total of 7523 documents were retrieved from the Web of Science Core Collection database, divided into two document types: articles and reviews. Then use R's built-in utility Biblioshiny for publication growth trend analysis, network analysis, highly cited literature analysis, factorial analysis, trend topics analysis, emerging themes analysis. Publishing trends in digital literacy research were uncovered: most cited documents, leading contributing authors, most productive countries and institutions, and journals contributing most to the field of study. Collaborative Network for Digital Literacy Research: Country Collaboration, Author Collaboration, Co-occurrence Network. Highly cited papers, Main topics, hot topics in digital literacy research, and emerging topics under the impact of COVID-19 pandemic. The COVID-19 epidemic is not over yet, and emerging themes related to digital literacy will continue to emerge, requiring further follow-up research. This article comprehensively summarizes digital literacy research work from 1990 to 2022. The publication trends, collaborative networks, topic evolution over the years, and emerging themes during COVID-19 pandemic. Nevertheless, although the research has a significant information and thus contribution to knowledge, findings and discussions are fairly presented. Specifically, the conclusion should contain clear contributions to academia and practitioners. Thus, more is needed to meet the quality standards of the respected journal. In addition, the references parts should be revised carefully by following the journal’s style of referencing.
Author Response
Response to Reviewer 4 Comments
Point 1: Nevertheless, although the research has a significant information and thus contribution to knowledge, findings and discussions are fairly presented. Specifically, the conclusion should contain clear contributions to academia and practitioners. Thus, more is needed to meet the quality standards of the respected journal.
Response 1: Thank you for your suggestion. We have re-modified and improved the conclusion section as follows.
- Discussion and Conclusion
From the publication output and growth trend results, it can be seen that the number of digital literacy publications has increased year by year. Still, the growth rate of digital literacy publications has risen significantly in recent years. Due to the onset of the pandemic and the shift to digital platforms as online education and working from home become the new normal, we expect more research in this area. From the network analysis, we can find that European and American countries have more cooperation; the cooperation among researchers is also relatively close. Digital literacy research can be roughly divided into two clusters: education and technology. For the highly cited papers we can find that papers’ articles on digital literacy research reviews have not only research reviews but also practical investigations, which have both theoretical reference value and social practice significance.
From the factorial analysis, we can find that the main directions of existing research are: "ICT, computer self-efficacy and gender gap", "Digital divide in Internet use, health literacy, health management", " Social media usage behavior of children and minors", "The use of digital literacy in English learning", "Student classroom education" and "Digital literacy for teachers". We can find that various problems and scientific literacy faced by people in the information society are long-term hot spots in digital literacy research. In addition, telemedicine and COVID-19 have become research hotspots in the past two years.
COVID-19 has undoubtedly had an impact on digital literacy research. Under the influence of COVID-19, "eHealth literacy", "mHealth", "health literacy", "digital health literacy", "digital health", "fake news", and "online learning" have emerged as emerg-ing topics. The COVID-19 pandemic is not yet over, and these topics deserve continued attention and exploration.
In the context of COVID-19, research into digital literacy continues to evolve. In the future, we will further refine our re-search points, such as strengthening research on topics such as "electronic health literacy", "digital health literacy", and "digital health". We will also further expand the scope of data collection, not just the WOS database, and update analysis methods such as machine learning text clustering.
Point 2: In addition, the references parts should be revised carefully by following the journal’s style of referencing.
Response 2: Thank you for your suggestion. We modified the reference format of the journal sample.
Reviewer 5 Report
In my opinion, the study explores an interesting resaerch topic, but needs follwoing changes during revision.
1- Please provide the complete term for the first time you use an abbreviation.
2- Abstract: There is no contribution(s) to knowledge and no closing comment.
3- The paper should provide a good and up-to-date review of literature in this field.
4- The Current Introduction doesn't persuade me. I agree with the concept; however, the sources and literature review are weak.
5- Please provide justification for selection criteria of current sample
6- Method is needed to be thoroughly extended.
7- Do not repeat the entire phrase after defining an abbreviation.
8- Conclusion should be concise along with the suggestions.
9- The manuscript should be improved in terms of the usage of English.
10. I would like to suggest the author to find more comparative studies related to the topic and include those studies in the current study to pursue his idea more strongly.
Author Response
Response to Reviewer 5 Comments
Point 1: Please provide the complete term for the first time you use an abbreviation.
Response 1: Thank you for your suggestion. We have made a full effort on the first term, such as OECD, ICT, etc.
Point 2: Abstract: There is no contribution(s) to knowledge and no closing comment.
Response 2: Thanks for your suggestions, we have revised and rewritten the Abstract section according to the journal format and article content. Part of the content is as follows, please refer to the revised draft for specific revisions:
Abstract:Background: Digital literacy is an increasingly important research topic today. Digital advances are supporting and accelerating the achievement of sustainable development, while at the same time, inequalities resulting from uneven diffusion of new technologies are emerging, especially as the world faces one of the greatest public health crises of the century, the COVID-19 pandemic. The primary purpose of this paper is to conduct an in-depth analysis of scientific production using bibliometrics, to understand the current research status of studies, and evaluate the pandemic's impact on digital literacy research. Methods: 7523 documents published between 1990 and 2022 were identified in the Web of Science database. This literature was analyzed using bibliometric research methods such as growth trend analysis, network analysis, highly cited literature analysis, factor analysis, and time series-based hot topic analysis. Additionally, a word cloud analysis of the keywords of digital literacy literature from 2020-2022 during the COVID-19 pandemic was performed. Results and Conclusions: Collaborative networks, key themes, and research trends in digital literacy research were identified, and the impact of the pandemic on digital literacy research was assessed.
Point 3: The paper should provide a good and up-to-date review of literature in this field.
Response 3: According to your suggestion, we have added the "Digital Literacy Research During the Covid-19 Pandemic" section.
Point 4: The Current Introduction doesn't persuade me. I agree with the concept; however, the sources and literature review are weak..
Response 4: According to your suggestion, we have added to the Introduction section. Highlight research issues and research goals. For specific content, please refer to the revised draft
Point 5: Please provide justification for selection criteria of current sample
Response 5: According to your suggestions ,we've added some references to support it. In addition, we also refer to the product introduction webpage of the WOS core database to prove that the WOS database can be used for research. At the end of the article, we also wrote that the scope of data search will be further expanded in future research, not just the WOS database.
Point 6: Method is needed to be thoroughly extended.
Response 6: At present, the commonly used method tools are VosViewer and Citespace. These two software are relatively mature, but they lack flexibility and functional diversity. The tools used in this article are based on the BIBLIOOMETRIX of the R package. Its literature measurement analysis functions are more extensive. In the future, we will join the LDA theme clustering and other machine learning methods to improve our research.
Point 7: Do not repeat the entire phrase after defining an abbreviation.
Response 7:. Thank you for your suggestion, we have performed grammar and spelling
Point 8: Conclusion should be concise along with the suggestions..
Response8 . With reference to your suggestions, we rewritten the Conclusion section
Point 9: The manuscript should be improved in terms of the usage of English..
Response9 . Thank you for your suggestion; we used English editing services to improve the usage of English.
Poin10: I would like to suggest the author to find more comparative studies related to the topic and include those studies in the current study to pursue his idea more strongly..
Response10 . Your suggestion is very welcome, and I also feel that the literature reviews were missing in the process of modification. Now they have been supplemented and modified.
Round 2
Reviewer 1 Report
The authors have improved the content and the quality of the research. I recommend the publication of the article after the authors use the format of the journal to revise the reference list.
Author Response
Point:
The authors have improved the content and the quality of the research. I recommend the publication of the article after the authors use the format of the journal to revise the reference list.
Response:
Thank you for your suggestions, which have greatly helped to improve the level of the article as well as my own writing skills. In addition, for the layout modification, we also sought expert editing services recommended by the journal and improved the logic and structure of the article, focusing on improving English grammar
Reviewer 2 Report
I consider authors are solving the suggestions.
Figure 7 is not clear the contribution.
Authors should check format style, empty pages should be deleted.
Author Response
Point1:Figure 7 is not clear the contribution.
Response1:
1. Thank you for your suggestion. The contribution of Figure 7 is indeed not obvious, so we have made the following changes:
Through the factorial analysis of the Keywords Plus, we can clearly identify two dimensions of digital literacy research, but it is impossible to clarify the sub-themes and the connections between them.To discover themes more intuitively, as shown in Figure 7, we converted the conceptual structure diagrams from the multiple correspondence analysis into a thematic tree diagram for analysis.
Point 2:Authors should check format style, empty pages should be deleted.
Response2:
We use the expert editing services recommended by the journal to improve the logic and structure of the article based on the improvement of English grammar.

Reviewer 3 Report
I appreciate that the Authors carefully read the reviewers' comments and tried to improve their paper. Thank you for the answers to my questions and for taking my suggestions into consideration as well. Unfortunately, I still have doubts about whether this paper is scientific enough. These doubts concern both the content and the language (English proofreading required).
For instance, in lines 34 to 58, the Authors added the research questions with justification. However, this justification is written more in a student than a scientist's style of writing.
There are still colloquial words used in the text, which is inappropriate for scientific papers (e.g. line 55 "budding researchers", line 90 "pieces of literature").
The whole paper rather delivers information than analysis. I do not see Authors connecting the dots and trying to find new research gaps or valuable reflections on the subject of digital literacy and its research. I simply expect more from the scientific paper. Therefore I would strongly recommend trying to work on the revised version of the paper in order to not leave the reader with the question "And so what?" at the end.
Author Response
Point1:The whole paper rather delivers information than analysis. I do not see Authors connecting the dots and trying to find new research gaps or valuable reflections on the subject of digital literacy and its research. I simply expect more from the scientific paper. Therefore I would strongly recommend trying to work on the revised version of the paper in order to not leave the reader with the question "And so what?" at the end.
Response1:
Thank you for your suggestions, your suggestions are very helpful to improve the level of the article. Not only articles, but also helpful to improve my own writing skills, thank you.
Based on your suggestion:
1. On the basis of the analysis and description of the article, the summary is strengthened, specifically:
1) Separate the discussion from the conclusion, and add some discussion content and problem thinking;
2) The Finding part, such as the Co-Occurrence Network part, explains and analyzes the results while describing the results.
2. In addition, we also used the expert editing service recommended by the journal to improve the logic and structure of the article on the basis of improving English grammar, which is also helpful to improve the level of the article.
Point2:
For instance, in lines 34 to 58, the Authors added the research questions with justification. However, this justification is written more in a student than a scientist's style of writing.
There are still colloquial words used in the text, which is inappropriate for scientific papers (e.g. line 55 "budding researchers", line 90 "pieces of literature").
Response2:
These two questions have been modified according to your suggestion

Reviewer 5 Report
my recommendation is publish
Author Response
Point:my recommendation is publish
Response:
Thank you for your previous suggestions, your suggestions are very helpful to improve the level of the article. Not only articles, but also helpful to improve my own writing skills, thank you.
In addition, for this revision, we also used the English expert editing service recommended by the journal to standardize the grammar and format of the article.

Round 3
Reviewer 3 Report
I do appreciate that the Authors tried to improve their paper accordingly to the remarks given by reviewers. And I do appreciate the work that has been done to prepare a bibliometric analysis.
I find the structure and the content more understandable. Unfortunately, there is still much to be done to find this paper valuable in contribution. The most important changes are needed in the Discussion part, as I do not see (probably not only me) the answer to the "So what?" question. I do understand that the paper is based on bibliometric analysis. But still, I have a problem with assessing what is its contribution to the current knowledge.
I would suggest rewriting the Discussion section to prove me wrong and highlight what is new and what is valuable in this paper.
Author Response
Suggestions:
I would suggest rewriting the Discussion section to prove me wrong and highlight what is new and what is valuable in this paper.
Response:
According to your suggestion, we have revised the article in the following aspects:
1. Rewrote the Discussion section and added the Findings section
2. In addition, the development part of the concept of digital literacy in the research review is transferred to the Introduction to enrich the background and introduction
3. According to the content modification, the Abstract part was rewritten
4. Modifications have been made to the English grammar and overall logic of the full text.

Round 4
Reviewer 3 Report
No further comments. Thank you for taking my suggestions into consideration.